# Molecular Characterization of Canine Parvovirus Variants CPV-2a and CPV-2c, Associated with Vaccinated Dogs at Libreville, Gabon

**DOI:** 10.3390/v15051169

**Published:** 2023-05-15

**Authors:** Gael Darren Maganga, Ingrid Labouba, Serda Zita Milendz Ikapi, Andriniaina Andy Nkili-Meyong, Audrey Michel Ngonga Dikongo, Larson Boundenga, Barthelemy Ngoubangoye, Christelle Memvie, Brice Serge Kumulungui

**Affiliations:** 1Centre Interdisciplinaire de Recherches Médicales de Franceville (CIRMF), Franceville BP 769, Gabon; 2Institut National Supérieur d’Agronomie et de Biotechnologies (INSAB), Université des Sciences et Techniques de Masuku (USTM), Franceville BP 913, Gabon; 3Cabinet Vétérinaire Veto Plus, Libreville BP 2619, Gabon

**Keywords:** canine parvovirus, vaccinated dogs, phylogeny, variants, Gabon

## Abstract

The first detection of canine parvovirus type-2 (CPV-2) was in the early 1970s, when it was known to cause severe gastroenteritis in dogs. However, it has evolved over the years into CPV-2a within 2 years, into CPV-2b after 14 years, into CPV-2c after 16 years and more recently CPV-2a-, 2b- and 2c-like variants reported in 2019, with a global distribution. Reports on the molecular epidemiology of this virus are missing in most African countries. The report of clinical cases among vaccinated dogs in Libreville in Gabon triggered the execution of this study. The objective of this study was to characterize circulating variants from dogs showing clinical signs suggestive of CPV that were examined by a veterinarian. A total of eight (8) fecal swab samples were collected, and all had positive PCR results. Sequencing, Blast analysis and assembly of two whole genomes and eight partial VP2 sequences were performed, and the sequences submitted to GenBank. Genetic characterization revealed the presence of CPV-2a and CPV-2c variants with predominance of the former. Phylogenetically, the Gabonese CPVs formed distinct groups similar to Zambian CPV-2c and Australian CPV-2a sequences. The antigenic variants CPV-2a and CPV-2c have not yet been reported in Central Africa. However, these CPV-2 variants circulate in young, vaccinated dogs in Gabon. These results suggest additional epidemiological and genomic studies are required in order to evaluate the occurrence of different CPV variants in Gabon and effectiveness of the commercial vaccines used against protoparvovirus in the country.

## 1. Introduction

Canine parvovirus disease is a highly contagious infection caused by the canine parvovirus type 2 (CPV-2), a small, non-enveloped, single-stranded DNA virus with 4–6 kb of nucleotides and classified within the family *Parvoviridae*, subfamily of *Parvovirinae* and the genus *Protoparvovirus* [1]. The CPV genome comprises two major ORFs, one encoding the two non-structural proteins (NS1 and NS2) and the other encoding the capsid proteins (VP1 and VP2) [2]. There is also a third protein, VP3, which is produced by a proteolytic processing of VP2 [3]. CPV-2 can cause acute and often fatal gastroenteritis manifested by vomiting and often bloody diarrhea associated with fever and anorexia, which affect domestic and wild carnivores. It is transmitted by direct or indirect contact through the fecal–oral route [4]. This disease has been reported to be more severe in puppies between 6 weeks and 6 months of age compared to their adult counterparts [5], although some reports have shown that acute enteritis can occur in dogs of all ages, breeds or sexes [6,7]. In puppies less than 3 months of age, the maternally derived antibodies (MDAs) play an important role in their protection but are also considered as the primary cause of vaccine failure against CPV-2 with high mortality and morbidity rates that can reach 50 and 90%, respectively [6,8,9,10]. Survival rates have been reported to be as high as 80–95% when cases are symptomatically treated early, but as low as 9.1% without treatment [7,11,12]. 

The original canine parvovirus type 2 (CPV-2) emerged in 1978, causing severe enteritis in dogs, and it was gradually replaced in the canine population by CPV-2 variants designated as CPV-2a, CPV-2b and CPV-2c with specific mutations in the VP2 capsid protein [6,13]. The 2a type strain has specific, important residues for typing at positions 87 (Met-Leu), 101 (Ile-Thr), 300 (Ala-Gly), 305 (Asp-Tyr), 555 (Val-Ile) and 426 (Asn-N for type 2a, Asp-D for type 2b and Glu-E for type 2c) and at position 297 (Ser-Ala for 2a/b) [2,14,15,16]. Antigenic variants are now identified worldwide [17]. Understanding mutations and the highly contagious nature of this virus has led to the use of vaccination as the main method for controlling the disease [17,18]. Vaccination is usually recommended at the age of six to eight weeks by the World Small Animal Veterinary Association (WSAVA) Guidelines for vaccination of dogs and cats [19]. There are several controversies on the need for new vaccines containing different strains of CPV-2 since cases of infected vaccinated dogs are observed [18,20,21,22,23]. Many authors point out that the occurrence of parvovirus infection cases in vaccinated dogs is mainly related to incorrect vaccination protocols rather than to vaccines that do not provide cross-protection against all circulating viruses. Vaccination too early, before 8 weeks of age, is ineffective because antibodies of maternal origin are able to interfere with the vaccine virus by preventing its replication and thus the development of protective immunity [22].

All three types of CPV-2 have been identified in dogs in Morocco, Nigeria, Tunisia and Egypt [3,21,24,25]. Only two subtypes, i.e., CPV-2a and CPV-2b, have been reported in southern Africa, particularly in Namibia, South Africa and Mozambique [26,27,28], and recently a first report of the predominance of CPV-2c in Zambia was published [20]. 

To date, no CPV variant has been reported in Central Africa, unlike in Northern, Southern and Western Africa, and most studies focus on the epidemiological profile and management of the disease [29]. In Gabon, reported clinical cases indicate the presence of the canine parvovirus disease, and the spread within the local canine population is of major concern. The detection and the molecular characterization of CPV-2 genotypes, as well as the identification of antigenic variants, have never been conducted in Gabon. In this study, we report the first cases of CPV-2 variants circulating in dogs at Libreville.

## 2. Materials and Methods

### 2.1. Sample Collection 

Veterinarians collected fecal samples in Libreville in 2019. A total of eight (8) swab samples were collected from domestic dogs of different breeds. Detailed clinical history of each dog was recorded (age, gender, vaccination status and symptoms). The collection method briefly consisted of inserting swab sticks into the dog’s anal cavity and turning them, then gently withdrawing them and transferring into tubes with stabilizing buffer. Swab samples were stored at −20 °C to maintain their integrity prior to laboratory testing at the Centre Interdisciplinaire de Recherches Médicales de Franceville (CIRMF). 

### 2.2. Clinical Findings and Dog Characteristics

In this study, samples were obtained from puppies and adults of different breeds (German shepherd, Basenji and Bichon frizzed dog) and ages varying from 5–12 months with suspected parvovirus. All dogs were vaccinated against canine parvovirus with different imported commercial vaccines but mainly with the live freeze-dried vaccine VAXIPET P (LAPROVET, Tours, France). This vaccine is composed of Parvovirus enteritidis canis. All animals presented diarrhea and other symptoms such as vomiting, weakness, loss of appetite and fever (Table 1). None of the sick dogs survived. The dogs died while in hospital. 

### 2.3. DNA Extraction and PCR Amplification

Viral DNA of the samples was extracted using QIAamp DNA Mini kit (Qiagen, Germantown, MD, USA) following the manufacturer’s instructions (www.qiagen.com/HB-0329, accessed on 10 April 2023). Briefly, 200 µL of each sample was incubated with 10 µL of proteinase K and 200 µL of lysis buffer at 56 °C for 15 min. Then, 200 µL of absolute ethanol was added in the lysate sample and mixed by pulse-vortexing for 15 s. Each sample was washed using two-wash buffer and centrifuged. The extracted nucleic acid was eluted with 100 µL of elution buffer, equilibrated to room temperature, quantified using a NanoVue spectrophotometer (Biochrom, Holliston, MA, USA) and then stored at −20 °C until its use.

The VP2 gene was amplified using primer pair 555forward/555reverse that amplifies a 583 bp fragment of the capsid protein [14]. The master mix and the protocol of the reaction were previously described [30]. The PCR reaction was performed in an Eppendorf Mastercycler nexus gradient machine (Eppendorf, Hamburg, Germany) with the following conditions: 95 °C for 10 min for initial denaturation, followed by 40 cycles of denaturation at 95 °C for 30 s, annealing at 55.5 °C for 30 s and extension at 72 °C for 3 min. Then, a final extension of 72 °C for 3 min was performed [30]. The obtained amplicons were sent to the Microsynth Seqlab laboratory in Göttingen, Germany, for sequencing using Sanger technology.

### 2.4. High-Throughput Sequencing

Positive samples obtained were sequenced using the Miseq machine (Illumina, San Diego, CA, USA). PCR products were purified directly from three distinct native biosamples using the QIAamp DNA mini kit (Qiagen, MD, USA), according to the manufacturer’s instructions. DNA extracts were quantified using a Qubit 2.0 fluorometer and fragmented by sonification using an M220 focused ultrasonicator (Covaris, Woburn, MA, USA). From these fragments, the libraries were prepared using the NEBNEXT Ultra^TM^II DNA library prep kit for Illumina (New England Biolabs, Ipswich, MA, USA) following the instructions of the manual. To limit the loss of genomic information, no selection size cleanup was performed during DNA library preparation. The quality control of the final DNA libraries was performed with the Agilent 2100 Bioanalyzer (Agilent Technologies, Santa Clara, CA, USA). A quantity of 8 µM of final DNA libraries was loaded for the Illumina sequencing, conducted using a Miseq Benchtop sequencer. A total of 300 cycles were achieved, and 150 nucleotide paired-end reads were obtained and assembled using SPAdes v 3.13.0 [31].

### 2.5. Phylogenetic Analysis

DNA sequences obtained were compared to those available in GenBank database using BLASTn (https://blast.ncbi.nlm.nih.gov/Blast.cgi, accessed on 8 November 2022) [32]. Sequences were edited and aligned with referenced sequences retrieved from the database which were selected based on genotype, geographic origin and similarity rates. Then, maximum-likelihood (ML) phylogenetic trees were inferred with MEGA 11 software using the neighbor-joining method and Tamura–Nei substitution model after muscle alignment, with 1000 bootstrap replicates [33,34]. In order to identify mutations in VP2 genes, the obtained sequences and their close relatives were translated into proteins. VP2 amino acid predictions were also aligned and analyzed in MEGA 11 for CPV variant typing [35]. The trees were modified using FigTree v1.4.4 software. 

## 3. Results

### 3.1. Molecular Diagnosis and Sequences Analysis

Sequence data generated in this study were deposited in the GenBank database under accession numbers from OP611195 to OP611204 and reported as Gabonese CPV strains. All samples submitted to CIRMF for parvovirus detection were confirmed positive with conventional PCR. After sequencing, ten (10) sequences were successfully recovered, including eight partial VP2 sequences with Sanger sequencing and two viral whole-genome sequences. Indeed, only two whole genomes were obtained; as for the other samples, the DNA concentrations were too low to perform high-throughput sequencing. Whole genomic sequences obtained in this study identified as cpv_gabon_G1 and cpv_gabon_G2 were in a single contig (i.e., had no gaps) with 4312 bp and 4671 bp, with G+C contents of 36.53 mol% and 36.12 mol%, respectively. The analysis of the partial fragment of VP2 gene sequences indicated that the majority of the sequences were highly similar (99.82–100% nucleotide identity) but showed changes in the amino acid at the position 426.

VP2 amino acid partial sequences analysis showed that the identified Gabon_Dog_M1 and Gabon_Dog_M2 had commonly found predicted amino acid changes in glutamic acid (Glu/E) at codon 426, which allowed its classification as antigenic variant CPV-2c (Table 2). In addition, the strain Gabon_Dog_M2 had a non-synonymous unique mutation I447M (Ile-Met). The other partial sequences Gabon_Dog_M3 to M8 had asparagine (Asn/N) at position 426 and valine at position 555, which is found in CPV-2 and CPV-2a.

### 3.2. Phylogenetic Analysis

Two phylogenetic trees were constructed, the first consisting of different partial short-length sequences of VP2 gene and the second with whole-genome sequences of Gabonese CPVs that were compared to selected closest variants circulating worldwide. A total of 51 nucleotide sequences were obtained from GenBank, including a feline panleukopenia virus (FPV) strain [36], added as an outgroup in each tree.

Concerning the ML phylogenetic tree of the partial VP2 genes, the analysis revealed a clustering of strains of the current study in two separate groups. Gabonese CPV-2c Dog_M1 and Dog_M2 strains were in a first group with other CPV-2c sequences from Zambia (LC409263) isolated in 2017, from Egypt (MK642273, MK642272) isolated in 2019 and from China (GU380305) isolated in 2009 (Figure 1). The Gabon_Dog_M1 CPV-2c (OP611197) strain was 100% identical at the nucleotide level to the Zambian strain (LC409263), while the percentage identity with Gabon_Dog_M2 CPV-2c strain was 99.82%. The other partial Gabonese sequences were in a second cluster with sequences of CPV-2a variants isolated from Australia in 2015 (KY242639, KY242640), New Zealand isolated in 2015 (KP881653), Hungary isolated in 2010 (EU815831) and even from China (for mink enteritis virus [36]) isolated in 2018 (MN747143). Meanwhile, the strains Gabonese_Dog_M3, M4, M5, M7 and M8 were identical (100% nucleotide identity). However, the Gabon_Dog_M6 sequence showed 99.81% nucleotide identity with the Gabon_Dog_M4 (OP611200) strain and 99.82% nucleotide identity with the others (Gabon_Dog_M3, M5, M7 and M8 sequences). The Gabon_Dog_M6 (OP611202) sequence stood out from others as indicated in Figure 2 and formed a completely distinct monophyletic clade. Phylogenetic results revealed that Gabonese strains Dog_M3 to M8 could be classified as a CPV-2a variant. 

Regarding the second ML phylogenetic tree shown in Figure 2, Gabonese whole-genome CPVs clustered with CPV-2c genomes from Thailand (MW589466), China (MH476592; MH476583), Vietnam (MT106228) and Egypt (OM937912) with high bootstrap support (100%). The cpv_gabon_G1 (OP611196) sequence formed a monophyletic cluster and was similar to two Chinese strains isolated in 2017 (MH476583 and MH476592), with 99.91% and 99.86% nucleotide identity, respectively, while the percent identity to the Thai strain (MW589466) was 99.88%. The cpv_gabon_G2 (OP611195) sequence shared a common ancestor with the strain from Thailand (MW589466) isolated in 2020 with a nucleotide identity percentage of 99.93%, while with the Chinese strains MH476583 and MH476592, this percentage was 99.87% and 99.81%, respectively.

## 4. Discussion

The findings of the current study indicate that CPV-2 is circulating in Gabon, and its spread has been increasingly reported in many other African countries in recent years [37,38]. All the eight samples collected were positive for CPV-2 infection as suspected clinically in sampled dogs. Canine parvovirus continues to be involved in several cases of gastroenteritis in domestic dogs [20]. The virus invades intestinal crypt cells and lymphoid tissues for its replication [39]. This leads to abdominal pain, bloody diarrhea, vomiting and extreme fatigue which were the most common symptoms observed in dogs in this study. These symptoms have also been reported in many other studies, with other symptoms including anorexia, hypovolemic shock, dehydration and severe immunosuppression causing sepsis, systematic inflammation and even fatal myocarditis [39,40,41]. During this study, more female dogs were found with CPV than males (6/8 cases against 2/8 cases). However, due to the small number of samples obtained, it would be difficult to draw conclusions on this result, although other studies found that males were more affected than females [29,42]. The small number of samples provided by veterinarians in this study could be justified by the fact that owners tend to only bring their dogs to clinics in case of severe disease [29]. In addition, in Gabon, there are veterinary structures in only three cities in the whole country. Cases of sick dogs had not been reported in these other cities during the period of this study. Thus, studies need to be conducted to determine the canine demography and epidemiological profile of CPV-2 infection in Gabon.

Here, we found that the CPV-2c and CPV-2a variants circulate in the capital city of Libreville with predominance of the latter. The results clearly indicate that the original CPV-2 strain and the CPV-2b variant were not found in this study, possibly due to the limited, non-representative sample size. The CPV-2a variant has been characterized and well reported in Africa, especially in Egypt, Morocco, Nigeria, South Africa, Tanzania, Tunisia, Zambia and in Namibia [3,20,21,24,25,27,43]. Since its emergence in 2000 in Italy, CPV-2c has been reported in North Africa and sub-Saharan Africa [3,14,20]. Apart from these studies, we provide here the first report of the molecular characterization of antigenic variants in Central Africa. These results should be taken into account in view of the data paucity from this region. 

Regarding the phylogenetic results, Gabonese CPV strains could have originated from Africa and Europe since their close relatives are from Zambia and Hungary. However, further phylogeographic analysis should be performed with a larger number of samples to better trace the origin of Gabonese CPV strains. Gabonese CPV-2a variants were found to be phylogenetically closely related to mink enteritis virus (MN747143) isolated from China in 2018, confirming their strong relationship as they share a common ancestor [44]. In addition, a single non-synonymous unique mutation I447M (Ile-Met) has been identified on the Gabon_Dog_M2 strain during alignment of amino-acid sequences. These results might support the evidence that the virus continues to evolve, considering the hypothesis that CPV-2c has a higher mutation rate than the other variants, which underlines the importance of ongoing surveillance programs to provide better understanding of the evolutionary dynamics of this virus in Gabon [20]. Other mutations could have been involved in Gabonese VP2 CPV-2 strains. Thus, the fragments of the VP2 gene containing the positions 87, 101, 267, 297, 300 and 305 need to be amplified for further amino acid mutations analysis. The clustering of Gabonese strains Dog_M1 (OP611197) and Dog_M2 (OP611198) with the other strains of CPV-2c variant in the phylogenetic tree confirms their classification into this antigenic type based on the Glutamic acid mutation at the position 426. For the strains Gabon_dog_M3 to M8 (OP611199-OP611204), despite the missing part of the VP2 fragment, the presence of asparagine at position 426 and valine at position 555 reinforces the results of the phylogenetic tree, showing a clustering with the CPV-2a variants. This allows them to be classified as CPV-2a strains from Gabon. Nevertheless, continual monitoring and molecular characterization of full-length VP2 genes from clinical samples and vaccine strains used is imperative to recognize mutations that can induce potential vaccine failures and mechanisms responsible for the evolution of CPV-2 [18,35,44,45].

In this study, Gabonese strains of CPV were isolated from vaccinated dogs. Findings raise concerns about a potential vaccine failure. In their practice, veterinarians in Libreville observe that in some dogs the vaccination protocol is not scrupulously respected. The booster shots are sometimes not given on time, which could increase the risk of vaccine failure. A great number of studies pointed out vaccine failure against CPV-2 variants and suggested that factors related to the vaccine itself or the health status of the animal could be the cause [21,29,46]. Possible explanations for these findings, among other factors, are poor vaccine quality, poor storage conditions, poor vaccine injection procedures by practicians, poor hygiene and low maintenance of the dogs, maternal derived antibodies, incomplete vaccination schedule and reversion of viral vaccines [22,47]. These results do not fit with the theory provided by Wilson et al. who reported in 2014 that vaccination with a CPV-2b-containing vaccine would induce cross-reactive antibody responses against other CPV-2 variants [18]. More recently, the scientific community has agreed that all CPV vaccines, regardless of the strain contained, when used well, are capable of protecting dogs against infections sustained by any variant, CPV-2a, CPV-2b or CPV-2c, even those absent in the vaccine [23,48]. However, vaccination against CPV-2 infection remains the main method to reduce the spread of the virus in canine population. A study described a non-zero prevalence, i.e., 81.5% (22/27), in vaccinated dogs in Argentina, showing that vaccines containing strains of antigenic types corresponding to variants circulating in the local dog population or polyvalent vaccines do not represent an alternative strategy to improve the effectiveness of prophylaxis against CPV [49]. There are inactivated vaccines and live vaccines sold worldwide to control CPV-2 infection [9]. Live vaccines are usually used for domestic dogs due to their long-term immunity and efficacy [44,50,51].

In the ongoing fight against CPV, due to the ability of the virus to mutate, giving rise to new more resistant and virulent subspecies, as well as its resistance in the environment and to common disinfectants and the difficulty eliminating it, three factors could help prevent the spread of CPV [52]. Firstly, daily sanitation of the environment would create an unfavorable environment for the survival of this virus. CPV is said to survive for up to one year in the environment and may be resistant to common detergent and disinfectants and extreme temperatures [49,53]. However, the virus could be sensitive or killed with diluted bleach if present on solid surfaces [53]. Secondly, observing a quarantine period would assist the elimination of the virus because CPV-2 is highly contagious. Infected dogs must be hospitalized and isolated to avoid spreading the virus to healthy animals and to prevent co-infection cases for example by CPV-2a and CPV-2c that have already been reported [53,54,55,56]. Then, along with the use of improved diagnostic tests, vaccination and several treatments such as the use of antibiotics, fluid therapies and adjunctive treatments might be necessary [44]. A study showed that the hemagglutination test could be the preliminary diagnostic approach for this disease due to its less sophisticated nature, low financial cost, speed and high sensitivity compared to the conventional PCR [30].

## 5. Conclusions

This study confirmed the circulation of both CPV-2a and CPV-2c in vaccinated dogs in Libreville with predominance of the former. This is the first report of CPV-2 infection in Gabon and the first time CPV-2c has been reported in Central Africa. Further studies are needed throughout the country to establish the current epidemiological profile and prevalence of antigenic variants in domestic dogs in Gabon. A particular emphasis on genomic surveillance, prevention and education is needed to effectively control and fight this virus’s spread.

## Figures and Tables

**Figure 1 viruses-15-01169-f001:**
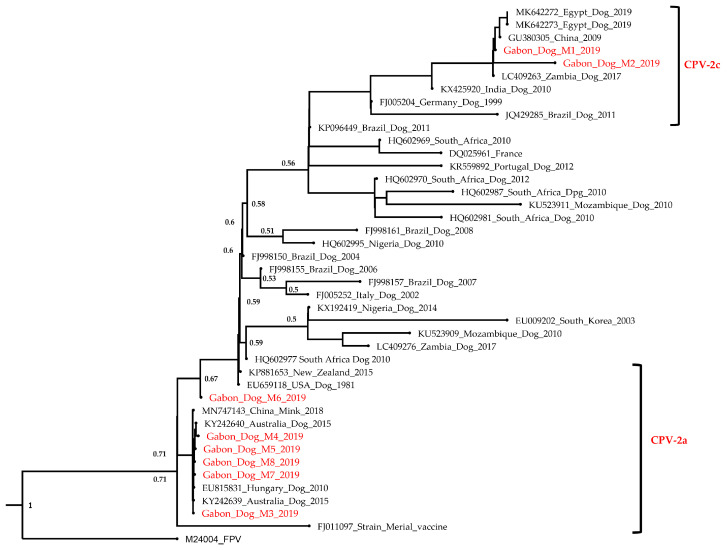
Phylogenetic tree based on partial nucleotide sequence VP2 gene of CPV-2 variants circulating in Gabon. This analysis involved 42 nucleotide sequences of canine parvovirus (CPV) obtained in this study and reference strains in the GenBank database. Sequences obtained in this study are highlighted in red. The tree was rooted with FPV (M24004), and bootstrap values lower than 50% were hidden.

**Figure 2 viruses-15-01169-f002:**
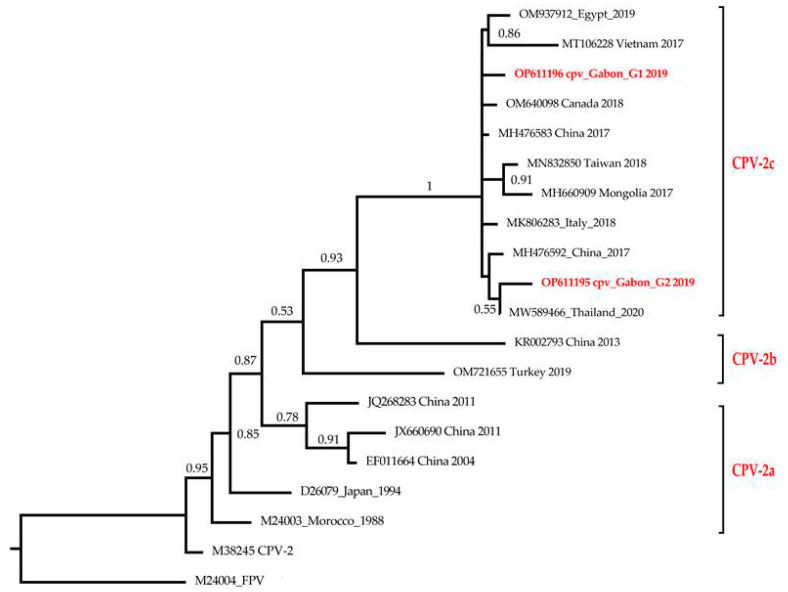
Phylogenetic tree of CPV-2 Gabonese whole genomes. Evolutionary analysis was conducted in MEGA 11 using the maximum-likelihood method and Tamura–Nei model, and taxa were arranged by input order. This analysis involved 20 nucleotide sequences, of which 17 were extracted from the GenBank database. The tree was rooted to FPV (M24004), and bootstrap values lower than 50% were hidden. The two whole-genome sequences obtained in this study are highlighted in red.

**Table 1 viruses-15-01169-t001:** Information concerning sampled dogs.

N	Age (Months)	Sex	Clinical Signs
1	6	F	Diarrhea; weakness; fever; vomiting
2	5	M	Bloody diarrhea; weakness; fever
3	8	F	Diarrhea; loss of appetite; fever; vomiting
4	6	M	Bloody diarrhea; weakness
5	8	F	Diarrhea; weakness; vomiting
6	12	F	Diarrhea; weakness
7	6	F	Diarrhea; weakness; vomiting
8	5	F	Bloody diarrhea, weakness, vomiting

**Table 2 viruses-15-01169-t002:** Amino acid variations in VP2 protein at key codons of Gabonese CPVs indicated by their GenBank accession numbers. Letters represent common amino acid abbreviations. The sequences of this study are the first eight species. Sequences OP611197 and OP611198 are CPV-2c variants with Glu/E/Glutamic acid changes at position 426. Sequences OP611197–OP6111204 are CPV-2a variants with Asn/N/asparagine changes at position 426.

Accession Numbers	80	87	93	101	103	297	300	305	323	426	555	564	568
OP611197	-	-	-	-	-	-	-	-	-	E	V	-	-
OP611198	-	-	-	-	-	-	-	-	-	E	V	-	-
OP611199	-	-	-	-	-	-	-	-	-	N	V	-	-
OP611200	-	-	-	-	-	-	-	-	-	N	V	-	-
OP611201	-	-	-	-	-	-	-	-	-	N	V	-	-
OP611202	-	-	-	-	-	-	-	-	-	N	V	-	-
OP611203	-	-	-	-	-	-	-	-	-	N	V	-	-
OP611204	-	-	-	-	-	-	-	-	-	N	V	-	-
CPV-2	R	M	N	I	V	S	A	D	N	N	V	S	G
CPV-2a	R	L	N	T	A	S	G	Y	N	N	I	S	G
CPV-2b	R	L	N	T	A	S	G	Y	N	D	V	S	G
CPV-2c	R	L	N	T	A	A	G	Y	N	E	V	S	G
FPV	K	M	K	I	V	S	A	D	D	N	V	N	A

## Data Availability

The sequences generated were deposited in GenBank (under accession numbers OP611195 to OP611204).

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
