# Peer review of "Molecular Characterization of Canine Parvovirus Variants CPV-2a and CPV-2c, Associated with Vaccinated Dogs at Libreville, Gabon"

_viruses, 2023, doi:10.3390/v15051169_

Round 1

Reviewer 1 Report

The manuscript “Molecular characterization of canine parvovirus variants  CPV-2a and CPV-2c, associated with vaccinated dogs at Libreville, Gabon” describes the genetic analysis of eight samples from vaccinated puppies with diarrhea and other clinical signs.  The article presents a very low number of samples and a very basic analysis. This must be worked on.

Concerns:

·        Recent studies, performing the phylogeny of all samples deposited in GenBank, verify that the samples form genotypes independently of the antigenic groups. This can be expected, since the antigenic groups are classified by mainly one site.

·        I'm not a native English speaker, but I noticed some spelling mistakes. The English must be checked.

·        In the methodology section, the dataset used must be described and presented. DNA substitution models must also be presented.

·        In my opinion, the description of the patients should be presented in the methodology section, unless there is a correlation with the obtained data.

·        The AA comparison should be performed by table, with a focus on seeing what changes occur for ancestral sequences. In the following, changes to local sequences can be analyzed. In the current format, we will lose a lot of evolutionary information.

·        In the discussion section, the authors comment on females being more susceptible to the virus in comparison to males. In the end, the authors comment that this can occur due to a low sample number. What seems more coherent to me. If there is no argument, this part can be removed.

·        In the discussion section, the authors comment that the origin of the virus in Gabon could be traced. This is correct, but it should be done with phylogeography. Unfortunately, the analyzes presented here have a very limited number of samples for this.

·        The discussion about the control of the virus in the environment is important, but it is disconnected with the results.

Author Response

Reviewer 1

Concerns:

  1. Reviewer: Recent studies, performing the phylogeny of all samples deposited in GenBank, verify that the samples form genotypes independently of the antigenic groups. This can be expected, since the antigenic groups are classified by mainly one site.

Response : We are totaly agree with the reviewer’s comment

  1. Reviewer: I'm not a native English speaker, but I noticed some spelling mistakes. The English must be checked.

Response : The english of the document has been checked as suggested by the reviewer.

  1. Reviewer: In the methodology section, the dataset used must be described and presented. DNA substitution models must also be presented.

Response: Tamura-Nei model was used and added to the methodology section in the manuscript

  1. Reviewer: In my opinion, the description of the patients should be presented in the methodology section, unless there is a correlation with the obtained data.

Response: The description of the patients was moved to the methodology section, as suggested by the reviewer.

  1. Reviewer: The AA comparison should be performed by table, with a focus on seeing what changes occur for ancestral sequences. In the following, changes to local sequences can be analyzed. In the current format, we will lose a lot of evolutionary information.

Response: As suggested by the reviewer, a table (table 2) was added instead of the Figure 1 showing the AA changes only of our Gabonese CPVs compared with referenced AA changes of CPV-2, CPV-2a, CPV-2b, CPV-2c and FPV

  1. Reviewer: In the discussion section, the authors comment on females being more susceptible to the virus in comparison to males. In the end, the authors comment that this can occur due to a low sample number. What seems more coherent to me. If there is no argument, this part can be removed.

Response: The following sentence has been deleted in the manuscript: “This might suggest that females are more susceptible to CPV infection than males“,

  1. Reviewer: In the discussion section, the authors comment that the origin of the virus in Gabon could be traced. This is correct, but it should be done with phylogeography. Unfortunately, the analyzes presented here have a very limited number of samples for this.

Response: We agree with the reviewer. We will be able to perform a more in-depth phylogeographic analysis by obtaining more samples in the future. This section has been reworded in the manuscript.

  1. Reviewer: The discussion about the control of the virus in the environment is important, but it is disconnected with the results.

Response: This part has been reworded

Reviewer 2 Report

The mansucript entitled “Molecular characterization of canine parvovirus variants CPV-2a and CPV-2c, associated with vaccinated dogs at Libreville, Gabon” by Maganga and co-authors characterized circulating variants from suspect CPV clinical sign dogs that were examined by a veterinarian. Authors have collected eight fecal swabs samples and performed the experimental and in-silico analysis. The manuscript is written well and all figures are relevant. I have a few queries that need to be answered prior to acceptance of the manuscript. I recommend a minor revision of the manuscript. Here are my suggestions/comments for the manuscript:-

1.    Manuscript needs to screen for typos and grammatical errors.

2.    Have authors confirmed the PCR amplification of any other method except sequencing ?

3.    Authors need to mention which software or tool was used for sequence analysis and comparison in Figure 1.

4.    It is not necessary to write all format of amino acid residues like three letter code, one letter code and full name in manuscript. For eg, in Figure 1 caption author written that “Sequences numbers 1, 2, 9, 14, 15, and 16 are CPVs-2c variants with Glu/E/Glutamic acid changes at the position 426. Sequences numbers 3-8, 10-13 are CPVs-2a variants with Asn/N/Asparagine changes at position 426.” Similarly in text also authors written “The other partial sequences Gabon_Dog_M3 to M8 had Asparagine (Asn/N) at position 426” and “Gabon_Dog_M2 had a commonly found predicted amino acid changes Glutamic acid (Glu/E) at codon 426”

Author Response

Reviewer 2

Here are my suggestions/comments for the manuscript:

  1. Reviewer : Manuscript needs to screen for typos and grammatical errors.

Response : Manuscript was screened for typos and grammatical errors as recommanded by the reviewer.

  1. Reviewer: Have authors confirmed the PCR amplification of any other method except sequencing ?

Response : Only the sequencing was used to confirm the PCR amplification as the primers used were specific.

  1. Reviewer: Authors need to mention which software or tool was used for sequence analysis and comparison in Figure 1.  

Response : MEGA11 software was used for sequence analysis and comparison. However, this figure 1 was finally replaced by a table based on the suggestions of a reviewer.

  1. Reviewer: It is not necessary to write all format of amino acid residues like three letter code, one letter code and full name in manuscript. For eg, in Figure 1 caption author written that “Sequences numbers 1, 2, 9, 14, 15, and 16 are CPVs-2c variants with Glu/E/Glutamic acid changes at the position 426. Sequences numbers 3-8, 10-13 are CPVs-2a variants with Asn/N/Asparagine changes at position 426.” Similarly in text also authors written “The other partial sequences Gabon_Dog_M3 to M8 had Asparagine (Asn/N) at position 426” and “Gabon_Dog_M2 had a commonly found predicted amino acid changes Glutamic acid (Glu/E) at codon 426”.

Response : The legend has been changed as suggested by the reviewer.

Reviewer 3 Report

Dear Authors,

The manuscript is interesting and well-designed; a grammatical revision is required throughout the text; for example, some articles, commas, adverbs, etc., must be included.

-       Throughout the text, check the conjugation of the verbs.

-       The paper needs to be checked carefully for typos and grammatical errors.

-       The scientific names and Latin names should be written in italics.

-       Keep the same font style and size throughout the manuscript and the figures.

-       At the reference list, some names of species are not italics.

-       The English version could be improved in some sentences. Please, ask for the support of a mother-tongue professor.

L18-L19. Changing the phrase to Chemicals that exhibit antimicrobial properties is one of the ingredients in cosmetics that ensure their durability and safety.

L55-L56. Eliminates the word at.

L61. Change of and use for new vaccines…

L67 – L69. I suggest changing the phrase to To date, no CPV variant has been reported in Central Africa except in Northern, Southern, and Western Africa, and most studies focus on the epidemiological profile and management of the disease [24].

L74 is the capital of Gabon.

L76-L77.  In the ethical approval section, after …observed in the dogs, please add the phrase: and use data for research purposes.

Is it necessary an approval document from the Committee on Bioethics?

L80 – L82. Changing the phrase to Veterinarians collected fecal samples in Libreville in 2019. A total of eight (8) swabs samples were collected from domestic dogs of different breeds.

L84 – L86. Add the reference.

L89. In DNA extraction and PCR amplification.

Add the reference for the DNA extraction protocol.

L94 – L95. Changing the phrase to The nucleic acid extracted was eluted with 100μL of elution buffer, quantified using NanoVue spectrophotometer (Biochrom, USA), and then stored at -20°C.

L105. The phrase corrected is: Positive samples obtained were sequenced using the MiSeq machine (Illumina, San Diego, CA, USA).

In the Phylogenetic analysis section, include the web address of all servers and databases.

L30-L34. Changing the phrase to In this study, samples were obtained from suspected puppies and adults of a different breed of the German shepherd, Basenji, and Bichon frizzed dog, aged 5 to 12 months. All dogs were vaccinated against canine parvovirus with different imported commercial vaccines. All animals presented diarrhea and other symptoms such as vomiting, weakness, loss of appetite, and fever (Table 1).

L135. In table 1, I suggest adding months in the age column.

L137-L47. The phrases should be improved, for example:

Sequence data generated in this study were deposited in the GenBank database under accession numbers from OP611195 to OP611204 and reported as Gabonese CPVs strains. All samples submitted to CIRMF for parvovirus detection were confirmed positive with conventional PCR. After sequencing, ten (10) sequences were successfully recovered, including eight partial VP2 sequences with the Sanger sequencing and two viral whole genome sequences.

L157. Figure legend. I suggest changing to Figure 1. Amino acid partial sequences comparison of capsid protein VP2 of Gabonese CPVs and close relatives indicated by their GenBank accession numbers. On the top from position 423 to 439 (left to right) and the bottom from position 544-560. Black frames highlight positions 426 and 555. Stars indicate identical amino acid positions. Letters represent standard amino acid abbreviations. The sequences of this study are the first eight species. Sequences 1, 2, 161, 9, 14, 15, and 16 are CPVs-2c variants with Glu/E/Glutamic acid changes at position 426. Sequences 3-8 and 10-13 are CPVs-2a variants with Asn/N/Asparagine changes at position 426.

L201. I suggest the figure legend. Figure 2. Phylogenetic tree based on partial nucleotide sequence VP2 gene of CPV-2 canine parvovirus variants circulating in Gabon. This analysis involved 42 nucleotide sequences of canine parvovirus (CPV) obtained in this study and reference strains in the GenBank database. Sequences obtained in this study are highlighted in red. The tree was rooted with FPV (M24004), and bootstrap values lower than 50% were hidden.

L207. I suggest the figure legend Figure 3. Phylogenetic tree of CPV-2 Gabonese whole genomes. Evolutionary analysis was conducted in MEGA 11, and taxa were arranged by input order. This analysis involved 20 nucleotide sequences, of which 17 were extracted from the GenBank database. The sequences obtained in this study are highlighted in red.

L300-L302. I suggest correcting the last phrase for: A particular emphasis on genomic surveillance, prevention, and education is needed to effectively control and fight this virus's spread.

Some significant points should be addressed before the manuscript can be considered for publication.

Author Response

Reviewer 3

Dear Authors,

The manuscript is interesting and well-designed; a grammatical revision is required throughout the text; for example, some articles, commas, adverbs, etc., must be included.

  1. Reviewer: Throughout the text, check the conjugation of the verbs.

Response: The conjugaison of the verbs has been checked as suggested by the reviewer.

  1. Reviewer: The paper needs to be checked carefully for typos and grammatical errors.

Response :  The paper has been checked as suggested by the reviewer.

  1. Reviewer: The scientific names and Latin names should be written in italics.

Response: We agree with the reviewer. The corrections were made in the manuscript.

  1. Reviewer: Keep the same font style and size throughout the manuscript and the figures.

Response: This was done as suggested by the reviewer.

  1. Reviewer: At the reference list, some names of species are not italics.

Response: This has been checked and corrected as suggested

  1. Reviewer: The English version could be improved in some sentences. Please, ask for the support of a mother-tongue professor.

Response: The English has been improved with the support of another English language specialist

  1. Reviewer: L18-L19. Changing the phrase to Chemicals that exhibit antimicrobial properties is one of the ingredients in cosmetics that ensure their durability and safety.

Response: We have difficulty understanding this comment from the reviewer. In lines 18-19 we do not find the correspondence with the suggestion made by the reviewer.

  1. Reviewer: L55-L56. Eliminates the word at.

Response: The word “at“ has been deleted as suggested.

  1. Reviewer: L61. Change of and use for new vaccines…

Response: This has been changed as suggested.

  1. Reviewer: L67 – L69. I suggest changing the phrase to To date, no CPV variant has been reported in Central Africa except in Northern, Southern, and Western Africa, and most studies focus on the epidemiological profile and management of the disease [24].

Response: The phrase has been reworded as suggested.

  1. Reviewer: L74 is the capital of Gabon.

Response: This has been deleted as suggested.

  1. Reviewer: L76-L77. In the ethical approval section, after …observed in the dogs, please add the phrase: and use data for research purposes.

Response: The sentence was added as suggested by the reviewer.

  1. Reviewer: Is it necessary an approval document from the Committee on Bioethics?

Response: In this case in Gabon it is not a necessity but this section is a requirement of the journal.

  1. Reviewer: L80 – L82. Changing the phrase to Veterinarians collected fecal samples in Libreville in 2019. A total of eight (8) swabs samples were collected from domestic dogs of different breeds.

Response: The phrase was reworded as suggested by the reviewer.

  1. Reviewer: L84 – L86. Add the reference.

Response: The term “clockwise“ has been deleted because there is no specific indication of which way the swab should be turned. This is a classic swabbing procedure that has been performed. Therefore, no specific reference has been added.

  1. Reviewer: L89. In DNA extraction and PCR amplification.

Add the reference for the DNA extraction protocol.

Response: The extraction protocol is contained in the QIAamp® DNA Mini Handbook available at: www.qiagen.com/HB-0329. This reference has been added in this section of the manuscript.

  1. Reviewer: L105. The phrase corrected is: Positive samples obtained were sequenced using the MiSeq machine (Illumina, San Diego, CA, USA).

Response: The phrase has been corrected as suggested.

  1. Reviewer: In the Phylogenetic analysis section, include the web address of all servers and databases.

Response: NCBI web address was added in this section as follow: https://blast.ncbi.nlm.nih.gov/Blast.cgi

  1. Reviewer: L30-L34. Changing the phrase to In this study, samples were obtained from suspected puppies and adults of a different breed of the German shepherd, Basenji, and Bichon frizzed dog, aged 5 to 12 months. All dogs were vaccinated against canine parvovirus with different imported commercial vaccines. All animals presented diarrhea and other symptoms such as vomiting, weakness, loss of appetite, and fever (Table 1).

Response: The paragraph was reworded as suggested.

  1. Reviewer: L135. In table 1, I suggest adding months in the age column.

Response: As suggested by reviewer, months was added in age column in the table 1.

  1. Reviewer: L137-L147. The phrases should be improved, for example:

Sequence data generated in this study were deposited in the GenBank database under accession numbers from OP611195 to OP611204 and reported as Gabonese CPVs strains. All samples submitted to CIRMF for parvovirus detection were confirmed positive with conventional PCR. After sequencing, ten (10) sequences were successfully recovered, including eight partial VP2 sequences with the Sanger sequencing and two viral whole genome sequences.

Response: The phrase has been reworded as suggested.

  1. Reviewer: L157. Figure legend. I suggest changing to Figure 1.Amino acid partial sequences comparison of capsid protein VP2 of Gabonese CPVs and close relatives indicated by their GenBank accession numbers. On the top from position 423 to 439 (left to right) and the bottom from position 544-560. Black frames highlight positions 426 and 555. Stars indicate identical amino acid positions. Letters represent standard amino acid abbreviations. The sequences of this study are the first eight species. Sequences 1, 2, 161, 9, 14, 15, and 16 are CPVs-2c variants with Glu/E/Glutamic acid changes at position 426. Sequences 3-8 and 10-13 are CPVs-2a variants with Asn/N/Asparagine changes at position 426.

Response: Figure 1 was changed by the table 1 as suggested by another reviewer, which completely changed the legend as follows: “Amino acid variations in VP2 protein at key codons of Gabonese CPVs indicated by their GenBank accession numbers. Letters represent common amino acid abbreviations. The sequences of this study are the first eight species. Sequences OP611197 and OP611198 are CPVs-2c variants with Glu/E/Glutamic acid changes at position 426. Sequences OP611197-OP6111204 are CPVs-2a variants with Asn/N/Asparagine changes at position 426“

  1. Reviewer: L201. I suggest the figure legend. Figure 2.Phylogenetic tree based on partial nucleotide sequence VP2 gene of CPV-2 canine parvovirus variants circulating in Gabon. This analysis involved 42 nucleotide sequences of canine parvovirus (CPV) obtained in this study and reference strains in the GenBank database. Sequences obtained in this study are highlighted in red. The tree was rooted with FPV (M24004), and bootstrap values lower than 50% were hidden.

Response: The legend of the figure 2 was reworded as suggested by the reviewer.

  1. Reviewer: L207. I suggest the figure legend Figure 3.Phylogenetic tree of CPV-2 Gabonese whole genomes. Evolutionary analysis was conducted in MEGA 11, and taxa were arranged by input order. This analysis involved 20 nucleotide sequences, of which 17 were extracted from the GenBank database. The sequences obtained in this study are highlighted in red.

Response: The legend of figure 3 was reworded as suggested.

  1. Reviewer: L300-L302. I suggest correcting the last phrase for: A particular emphasis on genomic surveillance, prevention, and education is needed to effectively control and fight this virus's spread.

Response: The last phrase has been corrected as suggested.

Reviewer 4 Report

This is an interesting paper which investigates the types of CPV circulating within Gabon, a previously into non-investigated area. The paper is generally quite well written, and contains some interesting information, however I have some major concerns with the paper.

The first is the small number of samples. Eight samples from a single city is not really enough for some of the conclusions which you make, and the two full genomes appear not to offer much different.

I am also concerned at the suggestion that they are vaccine failures. Apart from a throw away line in the methods (133-134) that the dogs were vaccinated, but no further details, including the type, when, if they produced an antibody response etc. It is also possible that vaccinated animals can shed the vaccine virus for a short period, but this has not been investigated, nor has the vaccine been sequenced. I think at very least, this section, and the title need to be toned down dramatically.

I also have some comments regarding the writing, which I have detailed below with suggested rewording.

More specific comments

Line 16- perhaps however rather than although may sound better?

Line 21perhaps variants from dogs showing clinical signs suggestive of CPV ….

Line 23- Sequencing, Blast analysis and assembly of two whole genomes and eight partial VP2 sequences were obtained and submitted to Genbank. … (reword)

Line 26- you can almost delete the ‘study reports the detection of CPV variants in the city of Libreville’ as you have almost already said it.

Line 30- genomic studies are required in order to …. (reword)

Line 39- space needed between made and of

Line 42- can delete virus after CPV

Line 43- associated with fever (reword)

Line 48- 3 months of age, CPV-2 is associated (delete the)

Line 54- The 2a type strain has specific important residues for typing at position ….. (reword)

Line 68- most studies focus on …. (reword)

Line 81- presenting with clinical signs …. (reword)

Line 94- Extracted nucleic acid was eluted … (reword)

Line 130- I am guessing that this should be ‘obtained from puppies with suspected parvovirus’?

Line 130- and adults of different breeds, including German Shepherd, Basenji and Bichon Frizzled dog, and ages varying from 5-12 months (reword)

Line 133- much more information is needed here- vaccine type, when vaccinated, did they produce a response etc? Without this, its almost impossible to make any judgement about if the vaccine virus is being shed, or if its an infection prior to immunity developing etc. Please also include vaccine type, and virus type in that vaccine. Also, did you see paperwork to confirm that they were vaccinated and been in contact with the pharma company who make it as a potential adverse event?

Line 144- were in a single contig (i.e. had no gaps) with 4,312bp and 4,671 bp respectively, and has G+C contents of 36.53% and 36.12% respectively (reword)

Line 147- changes in the amino acid … (reword)

Figure 1- I am not completely sure that these figures are needed but up to the authors

Line 168- obtained from Genbank … (reword)

Line 176- while the percentage identity with Gabon_dog_M2 CPV-2c strain was 99.82% (reword)

Line 178- CPV 2a variants isolated from … (reword)

Line 180- the strains were not very similar, they were identical, so please reword

Line 184 I would be tempted to have Gabon Dog M3, M5, M7 and M8 sequences in brackets to ease reading

Also in this section, having a number of mutations may be helpful along with the % identity?

Line 186- formed a completely distinct ….. (reword)

Line 188- could be classified as a CPV-21 …. (reword)

Line 212- delete ‘virus’

Line 222-Line 224- I think that your numbers are a bit too small to make these sort of comparisons

Line 231-232- again, I think that your numbers are a bit too small to make these claims

Line 241- I am not so familiar with this mink enteritis strain, but do please check it. It may be a wrongly labelled one on Genbank, or could be a CPV strain inoculated into a mink. MEV is usually more similar to FPV so this one seems a bit strange

Line 244- (Ile-Met) has been identified … (reword)

Line 246- 247- Not completely sure what you are saying here. I think you mean that CPV2c has a higher mutation rate than the other variants, but do please check and reference it

Line 259- full length VP2 genes from clinical … (reword)

Line 261- as detailed earlier, I don’t think that you can talk about vaccine failures with the data presented here. At maximum, you can talk about potential vaccine failures.

Line 273- what do you mean by a non-zero prevalence? Perhaps a value may be useful

Line 285- what do you mean by ‘et’ extreme ? Perhaps a typo?

Line 285-286- this is not a full sentence

Line 287-289- This is a good idea, but virus can be shed prior to showing clinical signs

Author Response

Reviewer 4

Comments and Suggestions for Authors

This is an interesting paper which investigates the types of CPV circulating within Gabon, a previously into non-investigated area. The paper is generally quite well written, and contains some interesting information, however I have some major concerns with the paper.

  1. Reviewer : The first is the small number of samples. Eight samples from a single city is not really enough for some of the conclusions which you make, and the two full genomes appear not to offer much different.

Response : We agree with the reviewer that the number of samples is rather small. It should be noted that this is due to the low number of veterinary practices in Libreville and the low adherence of the few available veterinary practices to the surveillance and sample collection system that had been put in place.

In Gabon, there are veterinary structures in only three cities in the entire country. And cases of sick dogs had not been reported in these other cities during the period of our study.

Although only two complete genomes were obtained, because for the other samples the DNA concentrations were too low to perform high-throughput sequencing, the analysis of these genomes confirms the existence of the CPV-2c antigenic variant which had never been reported previously in this country. It is for all these reasons, we have stated that Further studies are needed throughout the country to establish the current epidemiological profile and prevalence of antigenic variants in domestic dogs in Gabon.

With respect to the conclusions drawn, we have indeed gone back on some of them, for example, the conclusions on susceptibility to infection according to gender have been deleted due to the small number of samples.

Finally, despite the small number of samples, we believe that the results obtained deserve to be submitted for publication at least because of the extent of the analyses performed and because of the information provided concerning the genetic diversity of this virus and its geographical distribution.

  1. Reviewer : I am also concerned at the suggestion that they are vaccine failures. Apart from a throw away line in the methods (133-134) that the dogs were vaccinated, but no further details, including the type, when, if they produced an antibody response etc. It is also possible that vaccinated animals can shed the vaccine virus for a short period, but this has not been investigated, nor has the vaccine been sequenced. I think at very least, this section, and the title need to be toned down dramatically.

Response : The type of vaccine was indicated in the section “Clinical findings and dog’s characteristics“, i.e. a live freeze-dried vaccine VAXIPET P (LAPROVET, France). No other information was available except that the dogs had received their booster vaccination. Also, the vaccine has not been sequenced. Therefore, as suggested by the reviewer in one of his comments, we will talk about potential vaccine failure.

  1. Reviewer : I also have some comments regarding the writing, which I have detailed below with suggested rewording.

More specific comments

  1. Reviewer : Line 16- perhaps however rather than although may sound better?

Response : This has been corrected as suggested by the reviewer

  1. Reviewer : Line 21- perhaps variants from dogs showing clinical signs suggestive of CPV ….

Response : This has been corrected as suggested by the reviewer

  1. Reviewer : Line 23- Sequencing, Blast analysis and assembly of two whole genomes and eight partial VP2 sequences were obtained and submitted to Genbank. … (reword)

Response : The phrase was reworded as suggested by the reviewer

  1. Reviewer : Line 26- you can almost delete the ‘study reports the detection of CPV variants in the city of Libreville’ as you have almost already said it.

Response : This phrase was deleted as suggested

  1. Reviewer : Line 30- genomic studies are required in order to …. (reword)

Response : The phrase was reworded as suggested

  1. Reviewer : Line 39- space needed between made and of

Response : The space was added as suggested

  1. Reviewer : Line 42- can delete virus after CPV

Response : The word virus has been deleted

  1. Reviewer : Line 43- associated with fever (reword)

Response : The phrase was reworded as suggested

  1. Reviewer : Line 48- 3 months of age, CPV-2 is associated (delete the)

Response : This phrase has been modified following the suggestion of another reviewer  

  1. Reviewer : Line 54- The 2a type strain has specific important residues for typing at position ….. (reword)

Response : The phrase was reworded as suggested

  1. Reviewer : Line 68- most studies focus on …. (reword)

Response : The phrase was reworded as suggested

  1. Reviewer : Line 81- presenting with clinical signs …. (reword)

Response : The phrase was reworded as suggested by different reviewers

  1. Reviewer : Line 94- Extracted nucleic acid was eluted … (reword)

Response : The phrase was reworded as suggested

  1. Reviewer : Line 130- I am guessing that this should be ‘obtained from puppies with suspected parvovirus’?

Response : The phrase was reworded as suggested

  1. Reviewer : Line 130- and adults of different breeds, including German Shepherd, Basenji and Bichon Frizzled dog, and ages varying from 5-12 months (reword)

Response : The phrase was reworded as suggested

  1. Reviewer : Line 133- much more information is needed here- vaccine type, when vaccinated, did they produce a response etc? Without this, its almost impossible to make any judgement about if the vaccine virus is being shed, or if its an infection prior to immunity developing etc. Please also include vaccine type, and virus type in that vaccine. Also, did you see paperwork to confirm that they were vaccinated and been in contact with the pharma company who make it as a potential adverse event ?

Response : Regarding the vaccine, it is a live freeze-dried vaccine VAXIPET P (LAPROVET, France). The vaccine is composed of Parvovirus enteritidis canis. We did not ask to see the vaccination records. Therefore, we do not have any information on the vaccination period. We relied on the information provided by the veterinarians. These are dogs that are regularly monitored by the veterinarians with whom we have collaborated. And the observation that vaccinated dogs have parvovirosis is a general observation that has been noted for several years by all the veterinary clinicians in Libreville.  

  1. Reviewer : Line 144- were in a single contig (i.e. had no gaps) with 4,312bp and 4,671 bp respectively, and has G+C contents of 36.53% and 36.12% respectively (reword)

Response : The phrase was reworded as suggested

  1. Reviewer : Line 147- changes in the amino acid … (reword)

Response : The phrase was reworded as suggested by the reviewer

  1. Reviewer : Figure 1- I am not completely sure that these figures are needed but up to the authors

Response : The Fig1 was changed to tab1 as suggested by one reviewer

  1. Reviewer : Line 168- obtained from Genbank … (reword)

Response : The phrase was reworded as suggested

  1. Reviewer : Line 176- while the percentage identity with Gabon_dog_M2 CPV-2c strain was 99.82% (reword)

Response : The phrase was reworded as suggested

  1. Reviewer : Line 178- CPV 2a variants isolated from … (reword)

Response : The phrase was reworded as suggested

  1. Reviewer : Line 180- the strains were not very similar, they were identical, so please reword

Response : The phrase was reworded as follows : “the strains Gabonese_Dog_M3, M4, M5, M7 and M8 were identical between them“

  1. Reviewer : Line 184 I would be tempted to have Gabon Dog M3, M5, M7 and M8 sequences in brackets to ease reading

Response : This was done as suggested by the reviewer

  1. Reviewer : Also in this section, having a number of mutations may be helpful along with the % identity?

Response : We have understood the reviewer's suggestion. However, we have indicated the percentages of identity between each sequence or group of sequences, which is why we have not indicated the number of mutations.

  1. Reviewer : Line 186- formed a completely distinct ….. (reword)

Response : The phrase was reworded as suggested

  1. Reviewer : Line 188- could be classified as a CPV-21 …. (reword)

Response : This was reworded as suggested

  1. Reviewer : Line 212- delete ‘virus’

Response : The word “virus“ was deleted

  1. Reviewer : Line 222-Line 224- I think that your numbers are a bit too small to make these sort of comparisons

Response : Taking into account the reviewer's comment the sentence has been reworded as follows : “During this study, more female dogs were found with CPV than males (6/8 cases against 2/8 cases). However, due to the small number of samples obtained it would be difficult to draw conclusions on this result, although other studies found that males were more affected than females“

  1. Reviewer : Line 231-232- again, I think that your numbers are a bit too small to make these claims

Response : This sentence has been rewritten as follows : “ The results clearly indicate that original CPV-2 strain and CPV-2b variant were not found in this study possibly due to limited sample size that was not representative of the all.“

  1. Reviewer : Line 241- I am not so familiar with this mink enteritis strain, but do please check it. It may be a wrongly labelled one on Genbank, or could be a CPV strain inoculated into a mink. MEV is usually more similar to FPV so this one seems a bit strange.

Response : FPV (M24004) was used as outgroup instead of MEV and the phylogenetic tree result of CPV-2 Gabonese whole genome changed in the manuscript.

  1. Reviewer : Line 244- (Ile-Met) has been identified … (reword)

Response : This was corrected was reworded

  1. Reviewer : Line 246- 247- Not completely sure what you are saying here. I think you mean that CPV2c has a higher mutation rate than the other variants, but do please check and reference it

Response : This sentence has been reworded for clarity and a reference is provided

  1. Reviewer : Line 259- full length VP2 genes from clinical … (reword)

Response : This was reworded as suggested

  1. Reviewer : Line 261- as detailed earlier, I don’t think that you can talk about vaccine failures with the data presented here. At maximum, you can talk about potential vaccine failures

Response : This phrase was reworded following the reviewer's suggestions

  1. Reviewer : Line 273- what do you mean by a non-zero prevalence? Perhaps a value may be useful

Response : The exact value has been added, i.e. 81.5% (22/27 vaccinated dogs)

  1. Reviewer : Line 285- what do you mean by ‘et’ extreme ? Perhaps a typo?

Response : It is a typo. “et“ has been deleted

  1. Reviewer : Line 285-286- this is not a full sentence

Response : The sentence was reworded as follows : “ However, the virus could be sensitive or killed with diluted bleach if present on solid surfaces.“

  1. Reviewer : Line 287-289- This is a good idea, but virus can be shed prior to showing clinical signs

Response : We agree with the reviewer.

Reviewer 5 Report

The paper can be published after some corrections/adding/clarifications and a careful review of the English language.

For my complete comments see attached document (both corrected manuscript and comments in the same PDF file).

Author Response

Reviewer 5

Comments and Suggestions for Authors

The paper can be published after some corrections/adding/clarifications and a careful review of the English language.

  1. Reviewer : Page 1, line 20- Deleted “ ; “

Response : We deleted “ ; “

  1. Reviewer : Page 1, line 21- Deleted “sign“

Response : Following the recommendations of a reviewer, the sentence has been rewritten as follows : “The objective of this study was to characterize circulating variants from dogs showing clinical signs suggestive of CPV, that were examined by a veterinarian“.

  1. Reviewer : Page 1, line 23- Added “resulted“

Response : We added “resulted“ positive.

  1. Reviewer : Page 1, line 39- Space needed between made and of

Response : The space was added as suggested

  1. Reviewer : Page 1, line 43- Deleted “ ; “

Response : We deleted “ ; “

  1. Reviewer : Page 2, line 46- Space needed between “counterparts“ and the bracket

Response : The space was added as suggested

  1. Reviewer : Page 2, line 58- This comma divides subject and verb and must be removed

Response : The comma has been deleted as suggested

  1. Reviewer : Page 2, line 72- add a comma after “genotypes“

Response : The comma has been added as suggested

  1. Reviewer : Page 2, line 73- Deleted “the whole of“

Response : This was deleted as suggested

  1. Reviewer : Page 2, line 86- “the integrity“, Replace “the“ by “their“

Response : This was replaced by their as suggested

  1. Reviewer : Page 2, line 86- “was added“, replace “was“ by “were“

Response : This was replaced as suggested

  1. Reviewer : Page 3, line 94- Add “thus“ before “extracted“

Response : A correction to this sentence was suggested by a previous reviewer as follows : “ The extracted nucleic acid was eluted with 100µL…“

  1. Reviewer : Page 3, line 95- “biochrom“, reword “Biochrom“

Response : This was corrected as suggested

  1. Reviewer : Page 3, line 100- “of 95°C“, delete “of“ and replace by “ : “

Response : This was corrected as suggested

  1. Reviewer : Page 3, line 102- Add “was performed“

Response : The reworded as suggested

  1. Reviewer : not Illumina with capital letter and "a" at the end ?

Response : That's right. This has been corrected as suggested

  1. Reviewer : Page 3, line 114- Replace “technologies“ by “Technologies“

Response : This was corrected

  1. Reviewer : Page 3, line 115- Replace “were“ by “was“

Response : This was replaced

  1. Reviewer : Page 3, line 124- Replace “1000“ by “1,000“

Response : This was corrected

  1. Reviewer : Page 3, lines 130-132- Reword this sentence

Response : This sentence has been reworded following the suggestions of a previous reviewer. “frizzed“ has been replaced by “frise“. A comma has been added after “vomiting“

  1. Reviewer : Page 4, line 149- Replace “Amino“ by “amino“

Response : This was replaced by suggested

  1. Reviewer : Page 7, line 224- why two brackets?

Response : This has been corrected

  1. Reviewer : Page 7, line 230- “circulates“, delete the “s“

Response : This was corrected

  1. Reviewer : Page 8, line 242- Add a comma after “2018“

Response : A comma was added

  1. Reviewer : Page 8, line 244- Replace “have“ by “has“

Response : This was corrected

  1. Reviewer : Page 8, line 253- delete the comma after “tree“

Response : The comma was deleted

  1. Reviewer : Page 8, line 256- delete the semicolon after “555“ and replace by a comma

Response : The semicolon was deleted and replaced by a comma

  1. Reviewer : Page 8, line 285- delete “et“

Response : “et“ has been deleted

  1. Reviewer : Page 8, line 285- as long as the surfaces are first cleaned with hot soapy water or steam

Response : Totally agree with the reviewer

  1. Reviewer : Page 8, line 289- Replace “usually” by “for example“

Response : This was done as suggested

Round 2

Reviewer 4 Report

I think that the authors have done a good job with the manuscript revisions, and I extend my thanks to them for their hard work

Author Response

Thanks to the reviewer for his comments. Thank you for the reviews that helped to improve the quality of our manuscript.